# Effect of Fruit Volatiles from Native Host Plants on the Sexual Performance of *Anastrepha fraterculus* sp. 1 Males

**DOI:** 10.3390/insects14020188

**Published:** 2023-02-14

**Authors:** Guillermo Enrique Bachmann, Silvina Anahí Belliard, Francisco Devescovi, Ana Laura Nussenbaum, Patricia Carina Fernández, María Teresa Vera, María Josefina Ruiz, Diego Fernando Segura

**Affiliations:** 1Instituto de Genética “E.A. Favret”, INTA, GV-IABIMO, CONICET, Partido de Hurlingham B1686, Provincia de Buenos Aires, Argentina; 2Facultad de Agronomía, Universidad de Buenos Aires—CONICET, Ciudad Autónoma de Buenos Aires C1417, Argentina; 3Consejo Nacional de Investigaciones Científicas y Técnicas (CONICET), Ciudad Autónoma de Buenos Aires C1425FQB, Argentina; 4Facultad de Agronomía, Zootecnia y Veterinaria, Universidad Nacional de Tucumán, San Miguel de Tucumán T4000, Provincia de Tucumán, Argentina

**Keywords:** South American fruit fly, chemical ecology, semiochemicals, fruit volatile, sterile insect technique, Tephritidae

## Abstract

**Simple Summary:**

The South American fruit fly, *Anastrepha fraterculus*, is an important fruit pest. Males of this species are sexually stimulated by odors released by one of their main hosts, *Psidium guajava* (guava) fruit. Stimulated males release more pheromone and perform sexual displays more frequently than males that did not perceive fruit odors. This phenomenon was documented only for guava, a native host, whereas odors from other, exotic host species, such as mango, grapefruit and orange, do not trigger any stimulation in males. This might indicate that the male response is only triggered by native hosts, with which they have a common evolutionary history. Here we studied the effect of odors released by fruits of other four native host species of *A. fraterculus* sp. 1, namely *Eugenia myrcianthes*, *Juglans australis*, *Psidium cattleianum*, and *Acca sellowiana*, using guava as a positive control. Sexual displays and mating success were only affected by guava (as expected) and *P. cattleianum*. Interestingly, the two species belong to the same plant genus, which suggests a strong evolutionary relationship between *A. fraterculus* sp. 1 and, possibly plants of this genus. The present findings could help improving the sterile insect technique, an efficient pest control method widely used against fruit flies.

**Abstract:**

*Anastrepha fraterculus* sp.1 males are sexually stimulated by the aroma of fruit of its native host *Psidium guajava* (guava). Other hosts, which are exotic to *A. fraterculus*, do not enhance male sexual behavior. Here we evaluate the effects of fruit volatile exposure on male *A. fraterculus* sp. 1 sexual performance using other native hosts, under the hypothesis that male improvement derives from a common evolutionary history between *A. fraterculus* sp. 1 and its native hosts. Four species were evaluated: *Eugenia myrcianthes*, *Juglans australis*, *Psidium cattleianum*, and *Acca sellowiana*. Guava was used as a positive control. Males were exposed to fruit from 12:00 pm to 4:00 pm, from day 8 to day 11 post-emergence. On day 12, we evaluated their calling behavior and mating success. Both guava and *P. cattleianum* enhanced calling behavior. Mating success was enhanced only by guava and a trend was found for *P. cattleianum*. Interestingly, the two hosts belong to the Psidium genus. A volatile analysis is planned to identify the compounds responsible for this phenomenon. The other native fruits did not improve the sexual behavior of males. Implications of our findings in the management of *A. fraterculus* sp. 1 are discussed.

## 1. Introduction

Phytophagous insects have a close relationship with their host plants, which is often mediated by phytochemicals [1]. Besides the phytochemicals that are used as food by the insect (primary metabolites), plants produce compounds that affect the behavior and physiology of insects, either positively or negatively, without a direct effect on their nutrition (secondary metabolites). Phytophagous insects that mate on or near their host plant frequently use plant odors, a type of secondary metabolite, as a chemical cue to find mates [2]. Furthermore, these phytochemicals can influence the reproduction and sexual behavior of males and females, by inducing higher sex pheromone release rates, by affecting the composition of the pheromone, or by making it more attractive to the opposite sex [3,4,5]. In these species, a tight evolutionary relationship is expected between the host plant and its associated phytophagous insects, mediated by secondary metabolites.

The Tephritidae (Diptera) family comprises about 5000 species that inhabit tropical as well as subtropical and temperate zones throughout the globe [6]. Tephritid flies use healthy plant tissues, such as fruits, buds and flowers, from a wide range of species for larval development. Some of the members of this family oviposit in fruit; and since the list of host plants includes almost all commercial fruit species, many tephritid fruit flies are considered pests [7]. Due to their economic importance, fruit fly ecology has been the focus of many studies, particularly their interaction with host plants and how plants can influence their sexual behavior. (For a review, see Segura et al. [8].) In the *Bactrocera* genus, males of some species ingest methyl eugenol or raspberry ketone, two natural compounds found in many plants, including some of their hosts, and this triggers an increase in their sexual displays rate (calling behavior) and their mating success [9,10,11,12,13,14,15,16]. Likewise, *Ceratitis capitata* (Wiedemann) males increase their calling rate and mating success after being in the presence of host fruit, particularly *Citrus* fruit [17,18,19]. However, in this case, the mere exposure to fruit volatiles triggered the male response and the phytochemicals’ ingestion is not necessary. In the *Anastrepha* genus, males of two species, namely, *Anastrepha ludens* (Loew) and *Anastrepha serpentina* (Wiedemann), increase their mating success after exposure to grapefruit oil [20]. Despite the long list of examples where phytochemicals clearly, and positively, affect male fitness, the ecological, physiological, and evolutionary bases of this phenomenon are poorly understood [8,21,22].

*Anastrepha fraterculus* (Wiedemann), also known as “South American fruit fly”, is a complex of cryptic species that ranges from Mexico to Argentina [23,24,25,26,27] and is a serious fruit pest in many countries, including Brazil, Argentina and Peru [25,28,29]. The biology of *A. fraterculus* has been deeply studied (see Cladera et al. [30]), particularly its reproductive behavior (see reviews in Juárez et al. [31] and Vaníčková et al. [32]) and the male response to phytochemicals that could potentially provide a reproductive advantage, as previously shown in other tephritid fruit flies. Different studies have shown that the exposure of *A. fraterculus* from the Brazilian 1 morphotype (according to Hernandez-Ortiz et al. [33]), from now on *A. fraterculus* sp. 1, to volatiles of *Psidium guajava* L. (Myrtaceae) fruit, increased their courtship behavior, the amount of pheromone they release and their mating success [34,35,36]. Chemical characterization of the volatiles released by guava allowed us identifying potential compounds responsible for this phenomenon. Males exposed to a blend of seven compounds released by this fruit, namely, β-myrcene, (R)-limonene, (E)-β-ocimene, (E)-2-hexenal, α-humulene, ethyl hexanoate, and ethyl butanoate, showed an increased mating success [35], an effect later confirmed for limonene alone [37]. In turn, the rate of sexual calling of males increased after exposure to essential oil of guava fruit [38] and to (R)-limonene and (E/Z)-β-ocimene, compounds that are present in guava and guava essential oil [39]. Other sources of host fruit volatile compounds, such as mango fruit [34], orange essential oil and grapefruit essential oil [37], failed to sexually stimulate *A. fraterculus* sp. 1 males. Interestingly, volatiles from the essential oils of non-host species, such as lemon and a native shrub *Schinus polygama* (Cav.) Cabrera, induced an increase in male mating success [37], which suggests that these odor sources could be releasing compounds that are also present in guava (a host plant) and probably in similar abundances. This requires, nonetheless, further investigation.

The ultimate, evolutionary basis of the response of males to host plant compounds (or derived volatile sources) is not clear, but this phenomenon could be explained by, at least, three hypotheses. The first hypothesis (“sensory exploitation” hypothesis) is based on the concept of “sensory trap” [40,41], and postulates that males sequester volatile compounds released by guava and use them afterwards to lure females, exploiting a preexistent sensory bias in females [34]. For *A. fraterculus* sp.1, this would imply that males retain in their cuticle odors from guava that stimulate females [42]. The “synchronization” hypothesis proposes that males respond to guava volatiles to synchronize their behavior and invest resources in sexual activities when the host fruit (guava in this case) is available, ensuring receptive females and/or availability of hosts for their progeny [36,39]. In other words, host odors are read by males as a cue of increased chances to leave progeny. The last hypothesis, based on the “sexy son” hypothesis [43], proposes that females choose exposed males to mate because mating with such males would confer indirect benefits, as their sons will inherit their father’s foraging ability to locate and exploit host odors [44]. Under these three hypotheses, the response of males is expected particularly in the case of host species with which flies have a common evolutionary history. This is, actually, what has been found so far for *A. fraterculus* sp. 1, although a larger list of host species, particularly including native species, needs to be tested to confirm this.

The idea that *P. guajava*, being a native and important *A. fraterculus* sp. 1 host, stimulates males whereas exotic hosts fail to do so would require extending the list of host species analyzed. We are not aware of studies in which the fitness of *A. fraterculus* sp. 1 males in relation to other native host plant species has been evaluated. Therefore, here we aimed to analyze the effects of fruit volatile exposure on *A. fraterculus* sp. 1 male courtship behavior and mating success using native host species, other than guava, with which this fruit fly could have shared a common evolutionary history. Besides gaining basic knowledge on this interesting but poorly understood phenomenon, our study provides information of practical relevance. The sterile insect technique (SIT) is one of the main control methods against fruit fly pests [45]. The SIT involves the massive release of sterile insects over wide areas where the pest is a problem. There, the sterilized males and the wild fertile males compete to inseminate fertile females, which would then lay unviable eggs. Thus, the SIT development heavily relies on the sexual competitiveness of sterile males and any method that gives these males a competitive edge over their fertile counterparts would increase SIT efficacy [46]. For this reason, the results of our study are discussed in terms of potential SIT improvements.

## 2. Materials and Methods

Insects

Flies for the experiments were obtained from a colony established in 1997 with larvae recovered from guavas sampled in Tafí Viejo (26°43′25″ S, 65°16′43″ W), Tucumán, Argentina [47]. Rearing followed standard procedures using artificial diets, both for larvae and adults. Larval diet contains wheat germ, sugar, brewers’ yeast (CALSA, Tucumán, Argentina), and agar. Adult diet contains sugar, hydrolyzed yeast (MP Biomedicals, San Francisco, CA, USA), hydrolyzed corn (ARCOR, Tucumán, Argentina) (4:1:1 ratio), and vitamin E (Parafarm, Buenos Aires, Argentina) [47]. Within 24 h after adult emergence, flies were sorted by sex and transferred to 3 L glass flasks in groups of 30 individuals in Experiment 1, or groups of 5 males in Experiment 2 (see below). In all cases, females were fed with the standard adult diet described above and males were fed with sugar and brewer’s yeast (CALSA, Tucumán, Argentina) (3:1 ratio). This diet has already been proven to successfully enhance the sexual behavior of males when they are exposed to guava fruit volatiles [34,35,38,48]. Adults were kept under controlled environmental conditions (T: 25 ± 1 °C, RH: 60 ± 10%), and a 12L:12D light cycle, and were supplied with food and water ad libitum.

Fruits

Four native host species were included in the experiments. Three of them belonged to the Myrtaceae family: *Eugenia myrcianthes* (Nied.), *Psidium cattleianum* (Afzel. ex Sabine), *Acca sellowiana* (O. Berg); and one to the Juglandaceae family: *Juglans australis* (Griseb.) (Table 1). Furthermore, *P. guajava* was included in the experimental design as a positive control, based on our previous findings [34,35,48]. *Juglans australis* fruits were collected in a secondary forest near San Miguel de Tucumán (Argentina). *Psidium cattleianum* fruits were obtained from an experimental orchard near our laboratory, in Buenos Aires, while the rest of the fruits were collected from trees located on the sidewalks or in backyards near (<5 km) our lab (located in Hurlingham, Buenos Aires province, Argentina). Because different fruit species ripen in different periods of the year, the experiments were carried out during the ripening period of each fruit species, which sometimes overlapped, as indicated in Table 1. Nevertheless, all fruit species were harvested at a similar maturation degree (i.e., when the fruit reached its final size but was not ripe yet).

We only used fruits that had not been fumigated with pesticides. We chose fruits with no signs of fly infestation; however, we cannot rule out that some of the fruits used contained initial stages (eggs, L1, L2).

Experiments

In order to test the effect of the selected four native host species on the sexual behavior of *A. fraterculus* sp. 1 males, we carried out two experiments. All behavioral tests were performed under controlled laboratory conditions (T: 25 ± 1 °C, RH: 60 ± 10%).

### 2.1. Experiment 1. Effect of Native Host Fruit Volatiles on A. fraterculus sp. 1 Male Mating Success

In order to expose males of *A. fraterculus* sp. 1 to the volatiles of fruit, ten small pieces (ca., 1 g each) of the evaluated fruit were placed in a 20 mL container that was then covered with a mesh. This allowed us to expose males to the fruit volatiles but blocking them from contacting the fruit pieces (following Vera et al. [34] and Bachmann et al. [35]). The container with the fruit pieces was placed into the 3 L flask with 30 males for four hours (12:00 pm–4:00 pm) from day 8 to day 11 post-emergence. Control males and females were not exposed and maintained in separated environmental chambers under equal temperature and relative humidity conditions.

At day 12 post-emergence, males were evaluated in mating arenas in which each exposed male competed against a non-exposed male for access to a virgin, sexually mature female (12–14 days old). We used 1 L cylindrical containers made of transparent plastic as mating arenas, which have been extensively used as a valid experimental approach to evaluate female mate choice [34,35,36,48,49]. Exposed and non-exposed males were differentiated by a food dye (Laboratorios Fleibor, Buenos Aires, Argentina) that was added to the adult male diet and dyed the abdomen of the insects. Previous studies showed no detrimental effect of this kind of food dye on *A. fraterculus* sp. 1 behavior [31,37,49]. Colors were randomly assigned to different male treatments.

Just before the test at 8:00 am and under semi-darkness, an exposed male and a non-exposed male were placed in each experimental arena. Females were released after 15 min to give the males an acclimatization period and allow them to start the courtship displays (calling behavior, see below). Once all arenas were set up with the three flies (exposed and unexposed males and virgin female), natural light was gradually allowed in through a window, and the occurrence of mating pairs was continuously monitored for 2 h during the *A. fraterculus* sp. 1 natural period of mating activity (8–11 am) [50]. For each detected couple, the time at which copulation started and male color were recorded. Each mating pair was followed during the rest of the experiment (at 2–3 min time interval) to detect the separation of male and female, allowing to record the mating end time. Each mating pair was considered a replicate. Thus, for each replicate we recorded: (1) the successful male (exposed or non-exposed), (2) the latency to mate (time elapsed from the start of the experiment to the occurrence of mating) and (3) the mating duration (time elapsed between the mating pair was detected and the time the male and female disengaged). The number of replicates were 404, 544, 443, 434 and 609 for *E. myrcianthes*, *J. australis*, *P. cattleianum*, *A. sellowiana*, and *P. guajava*, respectively. Because of space and observers were limited, these replicates were carried out on different days (batches), which was considered in the statistical analysis (see below).

### 2.2. Experiment 2. Effect of Native Host Fruit Volatiles on A. fraterculus sp. 1 Male Sexual Calling Behavior

In order to quantify the effect of exposure on the frequency of male sexual displays, we recorded two behaviors related to *A. fraterculus* sp. 1 male courtship: wing fanning (bursts of wing fanning) and the extrusion of salivary glands (puffing of lateral pouches of expandable pleural abdominal cuticle, which is associated to the extrusion of the salivary glands) [50,51,52]. These behaviors have been extensively used as indicators of sexual displays in *A. fraterculus* sp. 1, as well as other Tephritidae fruit flies [8,35,38]. Wing fanning and salivary gland extrusion can occur simultaneously or not, so we recorded them separately as indicators of male sexual activity.

After emergence, groups of 5 males were placed in 3 L glass containers with water and food, where they were exposed to the fruits from day 8 to day 11 post-emergence, as it was described in Experiment 1. In this case, flies were kept in the same container and within the environmental chambers during the behavioral test; each container representing an experimental unit (replicate). At day 12, when the chamber lights turned on at 09:00 a.m., each container was observed 4 times at 30 min intervals. In each observation bout and container, we recorded the number of males that were observed performing wing fanning or exposing their salivary glands. The same number of replicates were carried out in parallel, but in separate chambers, for non-exposed males. The numbers of replicates included in this experiment were as follows: 18 for *E. myrcianthes*, 24 for *J. australis* and *P. cattleianum*, and 30 for *A. sellowiana* and *P. guajava*.

Data Analysis

All statistical analyses were carried out in R (v4.2.1) [53].

To estimate the effects of host exposure on male mating success, we performed a GLM model with binomial distribution [logit link function, *glmer* function from the *lme4* package [54]. For each host variant, the mating success was modeled as a function of treatment (exposed or non-exposed), and batches were included as a random factor. In addition, we estimated the effect of host exposure on latency to mate and mating duration. Given that the probability distributions of both variables were not normal, we used the *fitdist*, *descdist* and *gofstat* functions from the *fitdistrplus* package [55] to investigate alternative distributions. Latency did not fit to any tested density function. Therefore, the effects on this variable were evaluated using the nonparametric Kruskal–Wallis test with the *Kruskal.test* function from the *stats* package [53]. Mating duration fitted a gamma distribution, which was checked using the *simulateresiduals* function from the *DHARMa* package [56]. Therefore, we performed a GLM model with gamma distribution using the identity link function (*glm* function from the *stats* package) to estimate the effects of host exposure on mating duration.

To evaluate potential effects of fruit exposure on male sexual displays, we estimated the effect of exposure on two features of male calling behavior: wing fanning index (proportion of fanning males in each observation) and salivary glands exposure index (proportion of males exposing their salivary glands in each observation). For each of these male calling features, we averaged the four measurements per replicate, which were modeled as a function of treatment for each host. We performed a general linear model (GLM) using the *lmer* function from the *lme4* package [54]. Exposure was treated as a fixed factor, while batches were incorporated into the model as a random factor. We used the *leveneTest* function from the car package [57] and the *shapiro.test* function from the *stats* package [55] to assess for homoscedasticity and residual normality, respectively. The significance was evaluated with the *Anova* function from the *lmerTest* package [58].

## 3. Results

### 3.1. Experiment 1. Effect of Native Host Fruit Volatiles on A. fraterculus sp. 1 Male Mating Success

Mating success odds (the probability that a male mates divided by its complement) were on average 21% higher for males exposed to *P. guajava* than for non-exposed males (Z = 2.31, *p* = 0.021) (Figure 1). We found no effects of exposure on male mating success for any other host fruit species (*p* > 0.05). We did not find any effect of exposure on latency to mate and mating duration for any host fruit (Table 2).

### 3.2. Experiment 2. Effect of Native Host Fruit Volatiles on A. fraterculus sp. 1 Male Sexual Calling Behavior

Males exposed to *P. guajava* and *P. cattleianum* showed higher wing fanning index than non-exposed males (ChiSq1 = 8.35, *p* = 0.004 and ChiSq1 = 4.49, *p* = 0.034; respectively) (Figure 2A). In addition, males exposed to *P. cattleianum* showed a significantly higher salivary glands exposure index than non-exposed males (ChiSq1 = 15.7, *p* < 0.001) (Figure 2B). We found no effects of exposure on calling behavior for any other host fruit species (*p* > 0.05).

## 4. Discussion

On the basis of the well-known enhancement of *A. fraterculus* sp. 1 male sexual behavior after exposure to guava fruit volatiles [34,35,36,38,39], we hypothesized here that other native host fruits trigger a similar response on males of this important pest of South America. Therefore, we compared the effect of volatile exposure on male mating success and calling behavior of four native host fruits with that of *P. guajava*, and found that only *P. cattleianum*, the strawberry guava, presented a comparable effect to that of guava fruit volatiles. Volatiles from the other three native host fruits failed to enhance male mating success and calling rate. Mating duration and latency to mate were not affected by the volatiles of any fruit. These results support the idea of a strong evolutionary relationship between *A. fraterculus* sp. 1 and *P. guajava*, which has been postulated as a probable ancestral host of the species [59]. Furthermore, in the light of the present findings (i.e., a significant response to strawberry guava) this association may also involve other members of the same plant genus. Alternatively, flies might be responding to similar compounds being released by these closely related fruit species. These ideas should be, nonetheless, experimentally addressed in new studies.

Previous works generally show that volatiles of exotic host fruit did not affect the sexual behavior of *A. fraterculus* sp. 1 males [34,37], as opposed to what has been shown for guava, which is one of the most important native hosts of *A. fraterculus* sp. 1. As mentioned before, three hypotheses have been proposed, at least to our knowledge, to explain this association (namely, the “sensory exploitation” hypothesis, the “synchronization” hypothesis, and the “sexy son” hypothesis). Under these three hypotheses, a strong response to native hosts, with which *A. fraterculus* sp. 1 has a common evolutionary history, would be expected. Being *A. fraterculus* sp. 1 a polyphagous species, our prediction was, hence, that other native host fruits would induce an increase in male sexual signaling leading to increased sexual success. However, only *P. guajava* and *P. cattleianum* elicited this change in behavior. The fact that both species belong to the same genus suggests that *Psidium* genus might have a closer relationship with *A. fraterculus* sp. 1 than native host from other genera. Nonetheless, this idea should be tested with other *Psidium* species that are hosts of *A. fraterculus* sp. 1 (e.g., *Psidium araca* (Raddi), *Psidium guineense* (Sw.) and *Psidium kennedyanum* (Morong)).

At a functional level, the fact that fruit volatiles from *P. guajava* and *P. cattleianum* were able to sexually enhance *A. fraterculus* sp. 1 males might indicate the existence of shared compounds between the two host species. Because the two species belong to the same genus, a similar chemical composition of the fruit is expected, which is tightly associated with their volatile profile. It is known that a single compound can be enough to sexually stimulate males of other Tephritidae species. This is the case of *C. capitata* and α-copaene [60], and also *Bactrocera oleae* and α-pinene [61]. In other cases, exposure to natural sources such as fruits, fruit extracts or blends of specific compounds are necessary to enhance the males [18,19,20,35,37,38,39,62,63,64]. Particularly in *A. fraterculus* sp. 1, exposure to limonene enhances male sexual behavior [37,39], a response that was later increased by adding β-ocimene, in a specific ratio, to create a volatile blend of these two compounds [39]. However, other sources of volatiles containing these two compounds, such as orange and grapefruit essential oils [65], had no effect on males after fruit volatiles exposure [37]. The presence of detrimental compounds [39], enantiomers, different proportions of specific compounds that result in inadequate blends [66] or doses of compounds that might reach toxic levels [67] may explain the lack of effect to volatile sources containing limonene and ocimene. Actually, limonene at higher doses negatively affected the calling behavior of *A. fraterculus* sp. 1 [39]. Under this context, the two native host fruits reported here that enhanced the sexual behavior of males, that is *P. guajava* and *P. cattleianum*, should have aroma blends with a similar ratio of limonene and ocimene, which in turn should be different enough from the other fruits that did not affect male behavior. Hence, it would be important to address the presence and ratio of limonene and ocimene in these two fruit species and to compare this ratio with that of host species that did not affect male behavior and mating success. The lack of any of these two compounds or a different ratio in the other fruits evaluated can be considered supportive evidence.

An alternative explanation for the lack of effect of some native fruits would rely not so much on a natural phenomenon, but to experimental limitations of the methods we followed. These include differences between host species in terms of ripeness status when tested and also in the conditions under which males were exposed to the fruit. Firstly, during the fruit ripening process, volatile compounds released by the fruit change [68], and flies could be set to respond to a roughly specific combination of volatile compounds. As was previously mentioned, all fruit species were harvested at similar maturation degree. However, we cannot rule out the existence of some variation regarding the maturation degree among fruits of different species, which is a limitation that is difficult to overcome. Secondly, the conditions that were used during male exposure were defined for guava. As we repeated the experiments through the years, we have adjusted the protocol of exposure (mainly to reduce variability and workload). So, maybe other native fruit species need a somewhat different exposure protocol (amount of fruit, exposure time, the interval between exposure bouts, etc.) to trigger a positive effect on males. Belliard et al. [38] showed that the enhancing effect of guava volatiles depends on the presence and type of protein in the adult diet, which lead us to consider that other hosts could have eventually enhanced male sexual performance with a somewhat different diet formulation. These limitations could have introduced variations in our experiments. Again, studying the chemical profile of the fruit might help us to understand whether some of these host fruits could still be expected to have an effect similar to guava.

In general terms, the results obtained for mating success (Experiment 1) are consistent with results from the calling behavior tests (Experiment 2), across fruit species. Although *P. guajava* triggered a stronger response on male mating success than *P. cattleianum*—a significant effect was only observed for the former species—it increased both sexual calling behaviors: wing fanning and salivary gland exposure. In contrast, *P. guajava* only increased wing fanning. These two parameters are usually correlated [35,48,49,52], so we cannot explain why *P. guajava* exposure did not increase salivary gland exposure in males, especially when it improved their sexual success. Such a differential response to fruit exposure in different components of the male sexual courtship could affect female mating choice under natural conditions, so it should be studied in more detail, considering the importance of enhancing male sexual performance in the framework of the SIT.

A second point that is worth notice is the fact that even though *P. cattleianum* was the only fruit that increased the two behaviors used to estimate the sexual calling effort, it did not enhance the mating success, at least not as markedly as *P. guajava*. This can be seen as somewhat contradictory, since greater mating success is expected when sexual displays are more frequent in *A. fraterculus* sp. 1 [50]. The possibility of this result being due to low replication is easily discarded, since more than 400 mating arenas were examined from this host fruit. However, different batches of fruit were used for the mating success and calling behavior experiments, which may have caused such a discrepancy between the two behavioral tests. An effect of adaptation to the laboratory can also be discarded, as wild and laboratory flies responded in the same way to guava exposure [35]. Finally, mating tests not only involve an active engagement of males in calling but also reflect female choice, which may have been less selective due to unknown reasons, as they were always virgin, fed on a standard diet and used at the same age with all the tested native hosts. In any case, an increase in any of the indicators of sexual success (mating or calling) represents a promissory tool to use in the context of the SIT and an interesting research opportunity for exploring the physiological and genetic basis of such phenomena.

SIT success relies on sterile males’ sexual competitiveness against wild fertile males. In this work, we tested the effect of host fruit volatiles, alternative to *P. guajava*, on male calling behavior and mating success. The fruits chosen for these experiments were all hosts of *A. fraterculus* sp. 1 and native to South America, as we hypothesized that native fruits are most likely to improve male sexual performance. Apart from *P. guajava*, only one of the novel host fruits evaluated here (namely *P. cattleianum*, the strawberry guava) was successful at improving sexual behaviors commonly used as proxies to estimate male sexual success (i.e., calling behavior and mating success). These findings should be further validated under more realistic conditions, such as those field enclosures used to assess sterile male mating competitiveness [69] where sterilized males compete with wild fertile males for access to virgin wild females. As mentioned above, chemical characterization of the volatiles released by *P. guajava* and *P. cattleianum* might allow the elimination of variability from the behavioral tests by exposing males to a constant mixture of compounds, in specific ratios, instead of fruit collected from different trees and with varying ripening degrees.

## 5. Conclusions

Exposure to fruit volatiles of two native host species of *A. fraterculus* sp. 1 stimulated male sexual behavior.The two plant species that triggered this response belong to the *Psidium* genus. This phenomenon could result from an evolutionary association between *A. fraterculus* sp. 1 and host plants of this genus, or similarity in the aroma of its fruit, or both.Fruit volatiles from the other three native hosts did not enhance male mating success or calling behavior, which shows that males do not respond to any native host fruit, even though they are important (i.e., heavily infested) hosts in nature.

## Figures and Tables

**Figure 1 insects-14-00188-f001:**
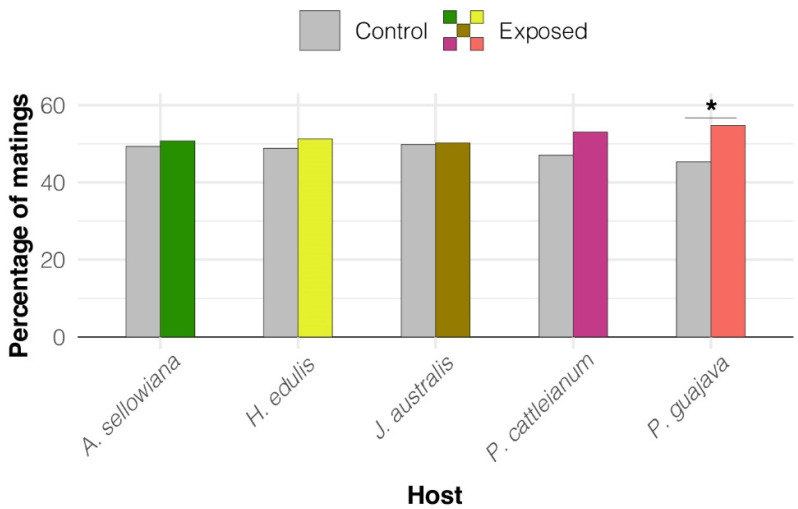
Effect of fruit volatile exposure on *Anastrepha fraterculus* sp. 1 male mating success. The figure shows the percentage of matings achieved by exposed and non-exposed (control) males for each native host fruit species. Asterisk indicates significant differences between exposed and non-exposed males (*p* = 0.021).

**Figure 2 insects-14-00188-f002:**
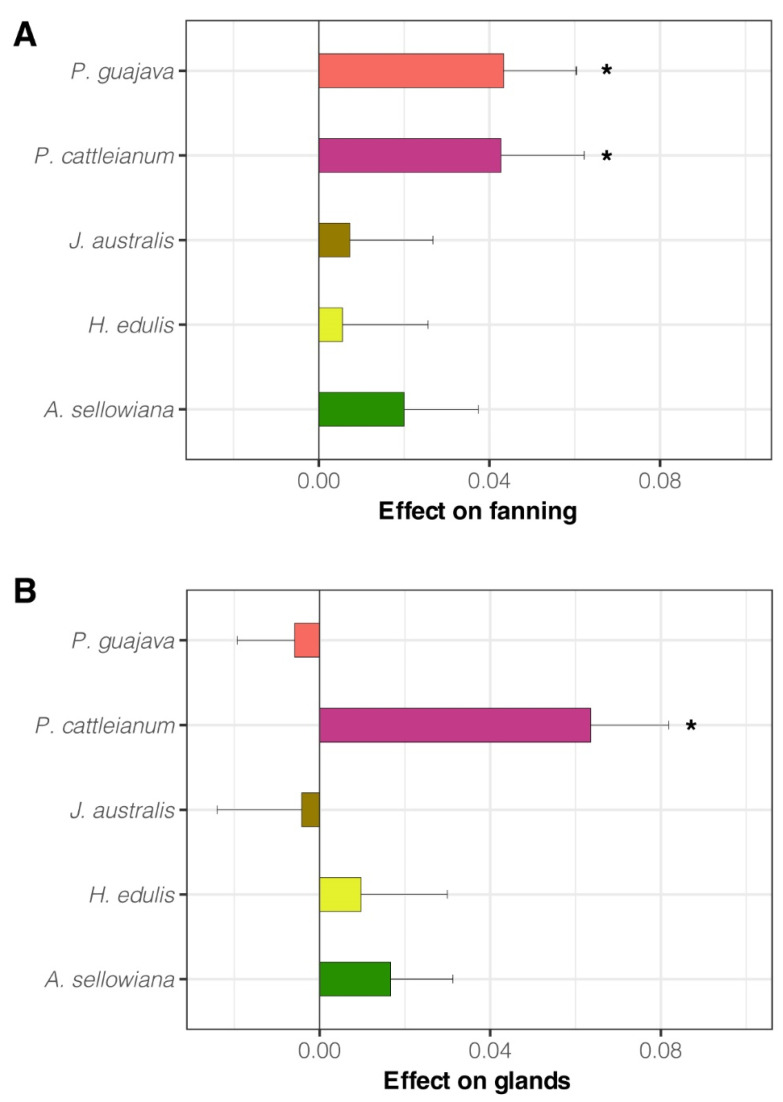
Effect of fruit volatile exposure on *Anastrepha fraterculus* sp. 1 male calling behavior, estimated through wing fanning (**A**) and salivary glands exposure (**B**). The figure shows the mean (±SE) effect on the male calling index (the difference between the calling index of exposed males and non-exposed males). Asterisks indicate significant differences between exposed and non-exposed males. Wing fanning: *Psidium guajava p* = 0.004, *Psidium cattleianum p* = 0.034; Salivary glands exposure: *P. cattleianum p* < 0.001.

**Table 1 insects-14-00188-t001:** Native host fruits included in the experiments.

Common Name	Scientific Name	Taxonomic Family	Distribution in Argentina	Fruits Ripening
Ubajay	*Eugenia myrcianthes*	Myrtaceae	Northeast	November–January
Argentine walnut (Nogal criollo)	*Juglans australis*	Juglandaceae	Northwest	December–February
Strawberry guava (Arazá rojo)	*Psidium cattleianum*	Myrtaceae	Northeast	February–March
Feijoa (Falso guayabo)	*Acca sellowiana*	Myrtaceae	North and central east	March–May
Guava (Guayaba)	*Psidium guajava*	Myrtaceae	Northeast	March–May

Native host fruits used to test the effect of fruit volatiles on *Anastrepha fraterculus* sp. 1 male sexual behavior. Fruits were obtained from experimental orchards and trees growing on sidewalks and backyards in Argentina.

**Table 2 insects-14-00188-t002:** Latency to mate and mating duration.

Host	Treatment	Latency			Duration		
		Mean ± SE	ChiSq	*p* Value	Mean ± SE	ChiSq	*p* Value
*Acca sellowiana*	Exposed	7.46 ± 0.56	0.036	0.850	65.50 ± 1.35	0.00004	0.570
	Non-exposed	8.24 ± 0.76			65.48 ± 1.38		
*Eugenia myrcianthes*	Exposed	9.00 ± 0.64	0.565	0.452	53.43 ± 1.37	0.067	0.796
	Non-exposed	9.55 ± 0.59			53.97 ± 1.58		
*Juglans australis*	Exposed	12.47 ± 0.77	0.850	0.355	70.98 ± 1.29	0.059	0.808
	Non-exposed	12.85 ± 0.70			71.41 ± 1.22		
*Psidium cattleianum*	Exposed	4.58 ± 0.47	0.087	0.768	61.08 ± 1.24	0.322	0.995
	Non-exposed	4.59 ± 0.51			60.09 ± 1.21		
*Psidium guajava*	Exposed	10.18 ± 0.65	0.0003	0.986	65.50 ± 1.35	0.507	0.477
	Non-exposed	9.91 ± 0.67			65.48 ± 1.38		

Latency to mate and mating duration (both in min) for exposed and non-exposed *Anastrepha fraterculus* sp. 1 males, for each native host fruit species.

## Data Availability

The data presented in this study are available upon request.

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
