# Peer review of "Effect of Fruit Volatiles from Native Host Plants on the Sexual Performance of Anastrepha fraterculus sp. 1 Males"

_insects, 2023, doi:10.3390/insects14020188_

Round 1

Reviewer 1 Report

The influence of plant metabolites on insect behavior is a very interesting and poorly known subject. Therefore, results of this study are of high scientific value.

General comments

“Anastrepha fraterculus” is a complex of cryptic species. The studied species “sp. 1” has not been described yet. It is a problem, because it is a pest and all information published on its biology should be easily accessed by using the unique and valid Latin name.

Could you please compile the formal description of this species and give it a valid name? For example, “Anastrepha (fratercula) una”? Otherwise all the information about the unnamed species would be sooner or later lost. One species is “sp. 1” for one researcher, and another species is “sp. 1” for another researcher. If you are not sure that it is a separate species, do not qualify it as a species. Call it “a population” or somehow like this.

Specific comments

Line 3. The word “Anastrepha” is a feminine noun. Therefore, according to International Code of Zoological Nomenclature, the name of species also should be feminine: Anastrepha fratercula. Since “fratercula” is not a species, but species group, this word should be placed into brackets: Anastrepha (fratercula) sp. 1

Line 24 The name "Hexachlamys edulis" is a junior synonym of Eugenia myrcianthes. Please, replace “Hexachlamys edulis” with “Eugenia myrcianthes” and check all other names of plants using International Plant Names Index https://www.ipni.org/. Only valid scientific names should be used.

Line 84. I am not sure, if the word “confer” is appropriate here.

Line 139. The behavior of insects cultivated in laboratory for a long time could significantly differ from the behavior of their native ancestors. Is it correct to study behavioral reactions on the laboratory colony initiated in 1997? Please, discuss this issue in the discussion section.

Line 167. Table 1 indicated that tested fruits ripe in different seasons. Please, write when the experiments were made. Were only ripe fruits used?

Figures 1 and 2  look vague.

Line 302. Strong reaction of these flies on P. guajava does not prove that it is their ancestral host plant. Ancestral host plant is not necessary the best host plant. For example, the ancestral host plant of Colorado potato beetle is Solanum rostratum. The best host, potato, did not occur in the small native range of this pest (Sonora).

 Please, add numbered Conclusions in the end of the manuscript.  

Author Response

Reviewer #1

The influence of plant metabolites on insect behavior is a very interesting and poorly known subject. Therefore, results of this study are of high scientific value.

General comments

“Anastrepha fraterculus” is a complex of cryptic species. The studied species “sp. 1” has not been described yet. It is a problem, because it is a pest and all information published on its biology should be easily accessed by using the  unique and valid Latin name.

Could you please compile the formal description of this species and give it a valid name? For example, “Anastrepha (fratercula) una”? Otherwise all the information about the unnamed species would be sooner or later lost. One species is “sp. 1” for one researcher, and another species is “sp. 1” for another researcher. If you are not sure that it is a separate species, do not qualify it as a species. Call it “a population” or somehow like this.

>> We appreciate all the comments made by Reviewer #1 and we agree with the Reviewer in that the Anastrepha fraterculus cryptic species complex should be solved by taxonomist soon, so each species can be defined with a unique Latin name. However, we are not taxonomists and makes poor sense we addressed this issue as part of a paper that deals with behavior and ecology. In order to avoid further misunderstanding, we used the most precise and accepted nomenclature available at the present time (Hernández-Hortiz et al. 2012). According to these authors, the morphotype we studied is the Brazilian 1 morphotype (sp. 1), which has been indicated in the manuscript. On top of that, we reported the geographic location where the flies used to establish the laboratory colony were sampled. We understand that it would be much simpler and direct to have a name for each species, but given the current taxonomic situation, we think it is better to name the species as we did in our previous works (as also other authors from Brazil working with this same morphotype are doing).

Specific comments

Line 3. The word “Anastrepha” is a feminine noun. Therefore, according to International Code of Zoological Nomenclature, the name of species also should be feminine: Anastrepha fratercula. Since “fratercula” is not a species, but species group, this word should be placed into brackets: Anastrepha (fratercula) sp. 1

>> As mentioned before, the taxonomy is outside our area of expertise and we do not consider that we have the capacity to rename species, especially when all the existing literature refers to this species as Anastrepha fraterculus. We believe that by changing the name now we would lose the possibility of this information being integrated with the preceding information published for this species under Anastrepha fraterculus. We thought it would be best to keep the name in this work and share the Reviewer’s comments with colleagues who are experts in tephritid taxonomy such as Vicente Hernández-Ortiz and Alan Norrbom.

Line 24 The name "Hexachlamys edulis" is a junior synonym of Eugenia myrcianthes. Please, replace “Hexachlamys edulis” with “Eugenia myrcianthes” and check all other names of plants using International Plant Names Index https://www.ipni.org/. Only valid scientific names should be used.

>> Changed accordingly.

.

Line 84. I am not sure, if the word “confer” is appropriate here.

>> Changed according.

Line 139. The behavior of insects cultivated in laboratory for a long time could significantly differ from the behavior of their native ancestors. Is it correct to study behavioral reactions on the laboratory colony initiated in 1997? Please, discuss this issue in the discussion section.

>> We understand the Reviewer concern, which a totally valid question. In our previous work (Bachmann et al. 2015), we compare the response of wild and lab males to guava exposure, using the same lab strain we used in the current work. We found no difference in the way males responded to the odors of fruit. On this basis, we feel confident using lab flies. We have included this information in the discussion (paragraph #6).

Line 167. Table 1 indicated that tested fruits ripe in different seasons. Please, write when the experiments were made. Were only ripe fruits used?

>> L167. Modified accordingly in section "Fruits" of Materials and Methods.

Figures 1 and 2  look vague.

>> Results are presented in standard figures for behavioral data. The Reviewer might find the figures as “vague” because the variable presented is the mean effect of exposure on the calling index. We think these figures suit perfectly the way the data has been analyzed and have explained in detail what the figure shows in the legend. No modifications were made.

Line 302. Strong reaction of these flies on P. guajava does not prove that it is their ancestral host plant. Ancestral host plant is not necessary the best host plant. For example, the ancestral host plant of Colorado potato beetle is Solanum rostratum. The best host, potato, did not occur in the small native range of this pest (Sonora).

>> Changed accordingly. With our results we do not prove that P. guajava is an ancestral host of A. fraterculus sp1.; we only commented that the results offer some support the proposed idea that this is the ancestral host of the species. We have explicitly commented on this in the revised version, following the Reviewer comment.

Please, add numbered Conclusions in the end of the manuscript. 

>>Changed accordingly. Conclusions were added to the revised version.

Reviewer 2 Report

I have no major issues with the manuscript by Bachmann et al., only a few minor ones to be addressed:

- line 92: correct to (E)-2-hexenal

- line 123: Maybe a different word to `improvement` is better here. Perhaps `male fitness`?

- line 314: Exp. 1 shows that only P. guajava confers sexual advantage, i.e. more successful mating attempts=increased male fitness, whereas P. cattleianum only modifies two aspects of courtship behaviour. From an evolutionary perspective, this indicates close links of the fly only with P. guajava. I agree with the discussion that follows about shared chemical profiles within Psidium and therefore potential similarities in their effects on male A. fraterculus.

Author Response

Reviewer #2

I have no major issues with the manuscript by Bachmann et al., only a few minor ones to be addressed:

>>We appreciate and value the comments made by Reviewer #2.

- line 92: correct to (E)-2-hexenal

>> Changed accordingly.

- line 123: Maybe a different word to `improvement` is better here. Perhaps `male fitness`?

>> Changed accordingly.

- line 314: Exp. 1 shows that only P. guajava confers sexual advantage, i.e. more successful mating attempts=increased male fitness, whereas P. cattleianum only modifies two aspects of courtship behaviour. From an evolutionary perspective, this indicates close links of the fly only with P. guajava. I agree with the discussion that follows about shared chemical profiles within Psidium and therefore potential similarities in their effects on male A. fraterculus.

>> We agree with Reviewer #2. We appreciate your comment and have commented this idea in the Discussion.